# Improvement of Luminescence Properties of Eulytite Single-Phase White Emitting Ca_3_Bi (PO_4_)_3_: Ce^3+^/Dy^3+^ Phosphor

**DOI:** 10.3390/molecules28134967

**Published:** 2023-06-24

**Authors:** Mengjiao Xu, Jiamin Liang, Luxiang Wang, Nannan Guo, Lili Ai

**Affiliations:** State Key Laboratory of Chemistry and Utilization of Carbon Based Energy Resources, College of Chemistry, Xinjiang University, Urumqi 830017, China; xmj_1117@163.com (M.X.); 19882996286@163.com (J.L.); guonan067@163.com (N.G.); ailili0709@163.com (L.A.)

**Keywords:** eulytite, phosphors, white light, energy transfer, single phase, thermal stability

## Abstract

To reduce the issue of tri-primary color reabsorption, a new approach for single-phase phosphors as light-emitting diodes (LEDs) has been recommended. The structures, morphology, photoluminescence, thermal stability, and luminescence mechanism of a variety of Ca_3_Bi (PO_4_)_3_ (CBPO): Ce^3+^/Dy^3+^ phosphors were investigated. XRD characterization showed that all CBPO samples were eulytite structures. Furthermore, the energy transfer process from Ce^3+^ to Dy^3+^ in CBPO is systematically investigated in this work, and the color of light can be adjusted by changing the ratio of doped ions. Under UV light, energy is transferred from Ce^3+^-Dy^3+^ mainly through quadrupole-quadrupole interactions in the CBPO host, and doping with different Dy^3+^ concentrations tunes the emission color from blue to white. The thermal stability of the CBPO: 0.04Ce^3+^, 0.08Dy^3+^ samples is outstanding, and the CIE coordinates of the samples after emission have little effect with temperature, while their emission intensity at 423 K is as strong as that at room temperature, reaching 90%. The above results indicate that this CBPO material has great potential as a white light phosphor under near-UV excitation at the optimized concentration of Ce^3+^ and Dy^3+^.

## 1. Introduction

Over the past few years, the use of phosphors in lighting, display, and anti-counterfeiting applications has been particularly significant and continues to expand. White LEDs are a new type of green solid-state light source developed in recent years that will replace the traditional light source (such as tungsten halogen lamps, incandescent lamps, etc.) because of their notable advantages such as environmental protection life and low power consumption [1,2,3]. The primary approach to fabricating WLEDs is to couple UV or n-UV chips (InGaN) with phosphors (Y_3_Al_5_O_12_). In general, phosphors are composed of an activator and a matrix material, sometimes a sensitizer, wherein rare earths or metals are frequently used as activator ions for matrix doping. High doping of single or multiple rare earth (RE) ions in a suitable single matrix produce red, green, blue as well as white light emission. In the lighting industry and display system, solid-state light source is an indispensable part of an LED phosphor; if the color adjustable phosphor is used to replace two single-color phosphors, it can avoid the color reabsorption and ratio control problems of a three-color phosphor. Therefore, single-matrix white phosphors have become a hot research topic in the field of phosphors for white LEDs.

Single-doped Dy^3+^ ions or co-doping of Dy^3+^/RE ions produce white light, thanks to the blue light generated by the ^4^F_9/2_-^6^H_15/2_ (465–495 nm) transition and the yellow light from the ^4^F_9/2_-^6^H_13/2_ (565–585 nm) transition. Nonetheless, the strictly forbidden 4f-4f transitions of Dy^3+^ cannot effectively absorb UV light, resulting in its poor luminous efficiency [4,5,6,7,8,9]. Among the many improvement strategies, energy transfer is a strategy to adjust the phosphor luminescence color, but also a feasible way to improve the phosphor luminescence efficiency [10]. Then it is crucial to find the appropriate sensitizer. In most of the RE ions, the Ce^3+^ ions are one of the ideal candidates for sensitizer, in which the 4f-5d of Ce^3+^ allows electric dipole transition and the 5d orbit is naked, allowing its emission range to vary from UV to visible region, and have been widely used to improve the luminous strength of these activator ions, such as Dy^3+^, Mn^2+^, Tb^3+^, Eu^3+^ ions, etc. [11,12,13,14,15,16]. Hence, by co-doping Ce^3+^ and Dy^3+^ into a suitable host, white light can be transferred through energy transfer between doping ions.

The selection of suitable substrate materials is essential to obtain energy-efficient phosphors. Orthophosphate, as a substrate for luminescent materials, has the advantages of stable physicochemical properties, excellent luminescent properties, low cost, and environmental friendliness, and it has received much attention and research from scholars in recent years. As an influential orthophosphate family, eulytite contains many inorganic substances with the formula A_3_B(PO_4_)_3_, where A is an alkaline earth, including Ca^2+^, Sr^2+^, Ba^2+^, et al., and B is a trivalent rare earth or Bi^3+^. Therefore, many studies have discussed the generation of novel high-efficiency luminescent materials with A_3_B(PO_4_)_3_ compound structures, such as a cubic symmetry Ca_3_Bi(PO_4_)_3_: Sm^3+^ orange phosphor obtained by Zou et al. [17]. Li et al. prepared Ba_3_Y(PO_4_)_3_: Ce^3+^/Tb^3+^, Tb^3+^/Eu^3+^ phosphors using an anion-cation substitution strategy, and the luminescence mechanism was elucidated in detail [18]. Wang’s group synthesized a new eulytite-type Pb_3_Bi(PO_4_)_3_: Eu^3+^ phosphor by high-temperature solid-phase reaction and investigated its structure, luminescence properties, and thermal stability [19]. Jiang et al. reported Sm^3+^ and Eu^3+^ co-doped Ca_3_Bi(PO_4_)_3_ red phosphors to study the photoluminescence properties and energy transfer of the phosphors [20]. Mukesh K. Sahu et al. reported the preparation of Ca_3_Bi(PO_4_)_3_: Er^3+^ materials with optical thermometric and photoluminescent properties using a co-precipitation method [21]. In recent work, Cao’s group reported Eu^3+^, Dy^3+^ co-doped Ca_3_Bi(PO_4_)_3_ single-phase phosphors and investigated the crystal structure, phase purity, and luminescence properties of the samples. The energy level diagrams of Dy^3+^ and Eu^3+^ ions were used to analyze the luminescence mechanism of the samples [22]. Out of all the types of eulytite, Ca_3_Bi(PO_4_)_3_ is the only kind of calcium eulytite synthesized at medium temperature, which has attracted wide attention. However, doping Ce^3+^ to enhance the luminescent properties of Dy^3+^ in Ca_3_Bi(PO_4_)_3_ host and applying it to LEDs has not been reported yet.

Based on the fact that energy transfer plays a crucial role in obtaining single-substrate white phosphors, the present work continues to try to obtain color-tunable phosphors by using energy transfer between dopant ions as a basis for obtaining white phosphors with a reasonable proportion of dopant ions. In our work, the host structure and luminescent properties of Ce^3+^/Dy^3+^ doped Ca_3_Bi(PO_4_)_3_ phosphors have been studied in detail. The prepared samples were characterized in series. The energy transfer process between Ce^3+^ and Dy^3+^ was analyzed. The phase structure, microscopic morphology, optimal doping amount, thermal stability, and luminescence of Ce^3+^/Dy^3+^ doped Ca_3_Bi(PO_4_)_3_ phosphors were analyzed by fluorescence spectroscopy, lifetime test, and transmission efficiency study characteristic. To sum up, the preparation of single matrix Ca_3_Bi(PO_4_)_3_: Ce^3+^/Dy^3+^ phosphors in n-UV base WLEDs have good application prospects.

## 2. Results and Discussion

### 2.1. Phase, Structure, and Morphology Analysis

The physical phase composition of the target product can be characterized by XRD. The XRD patterns of CBPO: 0.04Ce^3+^, CBPO: 0.08Dy^3+^, CBPO: 0.04Ce^3+^, 0.08Dy^3+^ samples, as shown in Figure 1a. All the doped and co-doped samples exhibit identical kinds of diffraction patterns indexed as (211), (220), (310), (321), (332), (422), (431), (530), (611), (620), (541), (631), (444), (721), and (732), basically consistent with JCPDS PDF card No. 85-2447. All samples are pure phase, and the structure of the matrix does not change with the addition of Ce^3+^/Dy^3+^ doping ions. The crystal structure of the CBPO compound is a cubic system, and the space group is I-43d (220), the lattice parameters are a = 9.977 Å, b = 9.977 Å, c = 9.977 Å, V = 993.18 Å3, Z = 4, see Figure 1b for details. CBPO has the structure type of eulytine, a three-dimensional network structure composed of octahedral CaO_6_/BiO_6_ and tetrahedral PO_4_. The Ca/Bi atoms are hexacoordinated, and the P atoms are tetracoordinated, which are located at the 16c and 12a sites, respectively. The O atoms occupy the 48e position and are randomly distributed in these two positions, thus causing the [PO_4_] tetrahedron to rotate disordered. Considering the ionic radii of Ce^3+^ (r = 1.01 Å) and Dy^3+^ (r = 0.912 Å) with Ca^2+^ (r = 1.00 Å) and Bi^3+^ (r = 1.03 Å), Ce^3+^/Dy^3+^ ions may replace Ca^2+^/Bi^3+^ ions. Ce^3+^ and Dy^3+^ may replace the site of Bi^3+^ due to the same valence state.

In order to verify whether the doping ions cause inhomogeneous phenomena such as elemental aggregation in the samples, the microscopic morphology and elemental distribution characterization of CBPO: 0.04Ce^3+^, 0.08Dy^3+^ were subsequently performed. Figure 1c shows the morphology of the CBPO: 0.04Ce^3+^, 0.08Dy^3+^ sample using SEM images. Since the target products were prepared by the high-temperature solid-phase method, it is noteworthy that the crystal particles of the prepared samples do not gather uniformly and have irregular shapes and sizes. To examine the homogeneity of the distribution of the sample, elemental mapping techniques are used to check the distribution of elements in the sample [see Figure 1c]. Experimental results showed that the elements Ca, Bi, P, O, Ce, and Dy are uniformly distributed in the samples, which means that the compositional homogeneity of the as-prepared phosphors is achieved.

### 2.2. Ce^3+^/Dy^3+^ Photoluminescence Properties Analysis

The luminescent behavior of phosphors can be characterized by FL. Figure 2 shows the photoluminescence excitation spectroscopy (PLE) and photoluminescence (PL) spectroscopy of the CBPO: 0.04Ce^3+^ phosphor. When excited at 320 nm, the CBPO: 0.04Ce^3+^ sample exhibits an asymmetrical emission band peak at around 370 nm in Figure 2a, which derives from the 4f^6^5d^1^-4f^7^ transitions of Ce^3+^. The PLE exhibits a narrow excitation band between 200–360 nm. The dependence of CBPO phosphor emission intensity on Ce^3+^ content is inset in Figure 2a. The matrix provides lattice sites for the dopant ions, which enter the lattice of the matrix, and the general concentration of dopant ions should not be lower than higher, as the high doping concentration will reduce the luminous efficiency of the luminescence process. The observed decrease in the luminescence intensity of the samples with increasing Ce^3+^ content is attributed to quenching, reaching a maximum at x = 0.04. Therefore, the optimal doping amount of Ce^3+^ was selected as 0.04 in this experiment, the concentration of Ce^3+^ was fixed, and Dy^3+^ was gradually introduced into subsequent codoped samples.

As depicted in Figure 2b, many sharp excitation peaks from 260 nm to 450 nm in the CBPO: 0.08Dy^3+^ sample of the PLE spectrum at 483 nm. The main excitation bands are attributed to the transitions from the ground state ^6^H_15/2_ to the following excited states of Dy^3+^: 325 nm (^4^F_5/2_, ^6^P_3/2_); 352 nm (^4^I_11/2_, ^6^P_7/2_); 365 nm (^4^M_19/2_, ^6^P_5/2_); 389 nm (^4^I_13/2_, ^4^F_7/2_); 426 nm (^4^G_11/2_, ^4^M_21/2_). A high luminescence peak at 352 nm was observed, assigned to the ^6^H_15/2–_^6^P_7/2_ transition, which was selected for further PL studies. For CBPO: 0.08Dy^3+^, there are mainly two transitions, ^4^F_9/2_-^6^H_15/2_ at 483 nm and transitions ^4^F_9/2_-^6^H_13/2_ at 575 nm, corresponding to blue and yellow light emission, respectively. The optical properties of a phosphor depend on its lattice structure. When in the blue emission band of the Dy^3+^ ion, it is at the center of a high symmetry inversion. In contrast, when the Dy^3+^ ion has a strong yellow emission intensity, it is at the center of a low symmetry inversion. From Figure 2b, It can see that Dy^3+^ is located at the highly symmetric site of the CBPO host lattice and emits more blue light at 483 nm. Comparing Figure 2a,b indicates that the resonance energy transfer of Ce^3+^ to Dy^3+^ in the CBPO host can be confirmed by the overlap of the PL spectrum of CBPO: 0.04Ce^3+^ and the PLE spectrum of CBPO: 0.08Dy^3+^. According to the formula given by Dexter [23,24]:(1)$$P_{SA}=2\pi /h\left|\langle S,A*\left|H_{SA}\right.\right.\left|S*,A\rangle \right.\left|{}^{2}\right.\int {}_{SA}g_{s}(E)g_{A}(E)dE$$
where *P_SA_* and *H_AS_* are the energy transfer rate and the interaction Hamiltonian, respectively; the matrix element indicates the interaction between the initial state *|S*, A>* and the final state *<S, A*|*. The integral represents the spectral overlap between the PL spectrum of Ce^3+^ and the PLE spectrum of Dy^3+^. From the observed spectral overlap between the PL band of Ce^3+^ and the PLE of Dy^3+^ and Equation (1), it can be concluded that resonance-type energy transfer is feasible from Ce^3+^ to Dy^3+^ in the CBPO host. Meanwhile, in Figure 2c, the PL spectra of CBPO: 0.04Ce^3+^, 0.08Dy^3+^ can be seen as the spectral combination of CBPO: 0.04Ce^3+^ and CBPO: 0.08Dy^3+^, which further confirms the above conclusion. The blue emission peaks at 370 nm and 483 nm and the yellow emission peak at 575 nm are due to the ^2^D_3/2_-^7^F_5/2_, ^4^F_9/2_-^6^H_15/2_, and ^4^F_9/2_-^6^H_13/2_ transitions of the Ce^3+^ and Dy^3+^ ions respectively. The excitation spectra of the samples CBPO: 0.04Ce^3+^, 0.08Dy^3+^ at 370 nm, 483 nm, and 575 nm all showed characteristic excitation peaks of Ce^3+^ at around 320 nm, and the position of the peaks remained unchanged with only a change in peak intensity.

To further explore the luminescence behavior among rare earth ions, then, the content of Ce^3+^ is fixed at 0.04, while the Dy^3+^ content varies from 0 to 0.16 in the co-doped samples. There are three emission bands of the CBPO: 0.04Ce^3+^, yDy^3+^ samples under 320 nm excitation wavelength [in Figure 3a]. Obviously, the intensity of the blue emission originating from Ce^3+^ declines progressively at the 370 nm peak. Meanwhile, the emission from Dy^3+^ enhances with the increase in y up to 0.08 and then decreases. This is mainly caused by the energy transfer behavior that occurs between ions in the matrix. In Figure 3c, the transition schematics of Ce^3+^ and Dy^3+^ further theoretically confirm the existence of the energy transfer process. The high phonon level of Ce^3+^ is excited to a low phonon level (^5^D_3/2_) by nonradiative relaxation. Energy transfer occurs between ^5^D_5/2_ and ^4^I_15/2_ of Ce^3+^, releasing to the ^4^F_9/2_ energy level of Dy^3+^. Due to the ^4^F_9/2_-^6^H_15/2_ and yellow ^4^F_9/2_-^6^H_15/2_ transitions, the ^4^F_9/2_ energy level rapidly enters the ground state nonradiatively and emits blue light. As shown in Figure 3c, the chromaticity coordinates (CIE) diagram is a tool that can be visually observed and judged by the human eye. Based on the energy transfer mechanism, the CIE coordinates of the series of samples No. 1–8 with (0.1603, 0.0207), (0.2239, 0.1517), (0.2398, 0.1891), (0.2599, 0.2346), (0.2686, 0.2562), (0.2801, 0.2797), (0.2928, 0.3053), and (0.2995, 0.3208). The emission color shifted from blue to white as the Dy^3+^ concentration (y) increased, as can be seen from the digital emission color photo in Figure 3c.

The fluorescence decay curves of the series of samples further verify the energy transfer between ions in Figure 4a. According to reported by Blasse and Grabmatier, the decay behavior of Ce^3+^ can be fitted by a first-order exponential the following equation [23,24,25]:(2)$$I=I_{0}\exp(-t/\tau )$$
where *τ* represents the luminescence decay lifetime, *I* represents the luminescence intensity at time t, and *I*_0_ represents the luminescence intensity at 0. For CBPO: 0.04Ce^3+^, yDy^3+^ (y = 0.00, 0.01, 0.02, 0.04, 0.08, 0.10, 0.12, 0.16) phosphors, the lifetime decreases with increasing Dy^3+^ concentrations, as 175.45 ns, 164.35 ns, 138.35 ns, 112.54 ns, 91.93 ns, 70.75 ns, 46.10 ns, and 29.00 ns, respectively. The energy transfer efficiency (η*_T_*) from Ce^3+^ to Dy^3+^ can be expressed by [26]:(3)$$\eta_{T}=1-\frac{\tau }{\tau_{o}}$$
where the *τ* represents the luminescent lifetime of Ce^3+^ with Dy^3+^ present, and *τ_o_* represents the luminescent lifetime of Ce^3+^ without Dy^3+^ present. The η*_T_* increases gradually with the doping contents of Dy^3+^, reaching 95% when y = 0.16 [in Figure 4b].

It is well known that the average critical distance (*Rc*) between Ce and Dy ions affects the type of energy transfer mechanism; when *Rc* is less than 5 Å, the exchange interaction dominates; when *Rc* is greater than 5 Å, the multipolar interaction is dominant. As suggested by Blasse, the *Rc* can be estimated by the following formula [23,27]:(4)$$R_{C}\approx 2{\left(\frac{3V}{4\pi X_{C}N}\right)}^{1/3}$$
where *V* stands for the volume of the unit cell, *X_C_* stands for the total concentration of Dy^3+^ and Ce^3+^, and *N* represents the number of cations. For the CBPO host, *X_C_* = 0.12, *V* = 993.18 Å3, and *N* = 4. The *Rc* was calculated to be about 15.81 Å. This means that exchange interactions do not play a role in the energy transfer process of CBPO: 0.04Ce^3+^, yDy^3+^. Therefore, the mechanism of energy transfer between Ce^3+^ and Dy^3+^ belongs to multipolar interactions. Based on the following relation can be obtained [28]:(5)$$\left(\frac{\eta_{0}}{\eta }\propto C^{n/3})≅(\frac{I_{S0}}{I_{S}}\propto C^{n/3}\right)$$
where *η*_0_ and *η* in the absence and presence of Dy^3+^ represent the luminescence quantum efficiency of Ce^3+^, respectively; its ratio can be approximately replaced by the ratio of the relevant luminescence intensity *I*_*S*0_/*I_S_*; *C* is the sum of the concentrations of doping ions; where *n* = 6, 8, and 10 correspond to dipole–dipole (d-d), dipole–quadrupole (d-q), and quadrupole–quadrupole (q-q) interactions, respectively. Figure 5 illustrates the relationship between *I*_*S*0_/*I_S_* and *C^n/3^*. Only when *n* = 10, the above results show that the energy transfer mechanism between Ce^3+^-Dy^3+^ is q- q because *I*_*S*0_/*I_S_* and *C^n/3^* are more linear.

### 2.3. Thermal Stability Analysis

For the practical application of a wLED device, the phosphor it contains will be subjected to higher temperatures in excess of 423 K (150 °C) when the device is electrically operated. Thermal stability is an essential parameter for measuring phosphors. Generally, the luminescence intensity of most phosphors will produce a temperature-quenching effect with a high working temperature [29,30]. Therefore, the best samples CBPO: 0.04Ce^3+^, 0.08Dy^3+^ were selected to study their PL spectra at different working temperatures (T = 173 K, 223 K, 273 K, 298 K, 323 K, 373 K, 423 K, 473 K), as shown in Figure 6a. When the working temperature is 423 K, the thermal stability of the samples CBPO: 0.04Ce^3+^, 0.08Dy^3+^ is excellent, and the emission intensity remains 90%. The coordinate configuration diagram can be used to explain the temperature quenching effect. As shown in Figure 6b, the green and yellow curves represent the ground and excited state, respectively. Point D is the intersection of the ground and excited state curves. At room temperature, electrons are first excited to their excited state under ultraviolet light. Most electrons return to their ground state by a radiative transition. With the increase in temperature, the energy makes the activator reach point D of the excited state and then return to the ground state 4F along the D-E-A route. No radiative transition results in decreased emission intensity and a significant temperature-quenching effect. Figure 6c, d show the CIE chromaticity coordinates of CBPO: 0.04Ce^3+^, 0.08Dy^3+^ sample at different temperatures. As the working temperature increases from 173 K to 423 K, the CIE coordinate values of the samples do not change significantly, which shows excellent color stability affected little by temperature (See Table 1). The CBPO: 0.04Ce^3+^, 0.08Dy^3+^ samples were subsequently re-tested for XRD and SEM after a working temperature of 473 K, as shown in Appendix A. The XRD data in Appendix A show that the sample retains the diffraction peak of CBPO after the working temperature of 473 K, in general agreement with its standard card JCPDS PDF 85-2447. As can be seen from the SEM images in Appendix A, the sample, after working at 473 K, remains consistent with the original sample in terms of its morphology, aggregation, and irregular shapes. It is further suggested that the sample exhibited good stability.

## 3. Materials and Methods

### 3.1. Preparation Process

The samples of Ca_3_Bi_1−x_(PO_4_)_3_: xCe^3+^ (x = 0.005, 0.01, 0.02, 0.04, 0.06, 0.08, 0.10, x represent molar concentration), and the content of Ce^3+^ is fixed at 0.04, the samples of Ca_3_Bi_0_._96 −y_(PO_4_)_3_ (CBPO): 0.04Ce^3+^, yDy^3+^ (y = 0.01, 0.02, 0.04, 0.08, 0.10, 0.12, 0.16, x and y represent molar concentration) samples were prepared by two-step calcination in a weak reducing atmosphere by a high-temperature solid-phase method. The raw materials are all analytical reagents, including CaCO_3_, Bi_2_O_3_, NH_4_H_2_PO_4_, CeO_2_, and Dy_2_O_3_. To obtain the target product, the raw material was first poured into an agate mortar and thoroughly ground, and then it was calcined in a tube furnace for two steps in a reducing atmosphere, including heating at 180 °C for 4 h and then heating at 1200 °C for 3 h. Finally, the target product is obtained after final cooling.

### 3.2. Characterizations

A scanning electron microscope (SEM, model: Hitachi S-4800) was used to describe the morphology and size of the phosphors with an attached energy dispersive spectroscopy (EDS) function. A powder X-ray diffractometer (XRD, model: Bruker D8) was used to characterize the phases of the samples. A fluorescence spectrometer (FL, model: Hitachi F-4500) was used to measure the phosphor’s fluorescence spectroscopy, and the excitation light source was a 150 W Xe lamp. HORIBA Jobin Yvon Fluorolog-3 was used to test the samples’ decay curve and thermal stability.

## 4. Conclusions

A series of CBPO: xCe^3+^ and CBPO: 0.04Ce^3+^, yDy^3+^ phosphors have been synthesized by solid-state reactions in a weak reducing atmosphere via secondary calcination. The optimal single-doped Ce^3+^ ion concentration of 0.04 was determined, and subsequently, the optimal double-doped Ce^3+^ and Dy^3+^ concentrations were determined at fixed altered concentrations. In the meantime, the PL of CBPO: 0.04Ce^3+^, 0.08Dy^3+^ samples showed blue emission peaks at 370 nm and 483 nm and yellow emission peaks at 575 nm, which were due to ^2^D_3/2_-^7^F_5/2_, ^4^F_9/2_-^6^H_15/2_, and ^4^F_9/2_-^6^H_13/2_ transitions of Ce^3+^ and Dy^3+^ ions, respectively. The experimental data and calculations show that the q-q mechanism is that the energy transfer from Ce^3+^ to Dy^3^+ is resonance type, the maximum η*_T_* is 95%, and the *Rc* calculated is 15.81 Å. CBPO: 0.04Ce^3+^, yDy^3+^ (y = 0.00, 0.01, 0.02, 0.04, 0.08, 0.10, 0.12, 0.16) phosphor changed its luminescence color from blue to white. The thermal stability of the CBPO: 0.04Ce^3+^, 0.08Dy^3+^ samples are outstanding, and their emission intensity is 90% as strong at 423 K as it is at room temperature. The white light can come true by tuning the ratio of Dy^3+^, and the addition of Ce^3+^ can simultaneously improve the luminescence performance of Dy^3+^, which indicates that CBPO: Ce^3+^, Dy^3+^ phosphors have great potential as single-phase white light phosphors excited by n-UV.

## Figures and Tables

**Figure 1 molecules-28-04967-f001:**
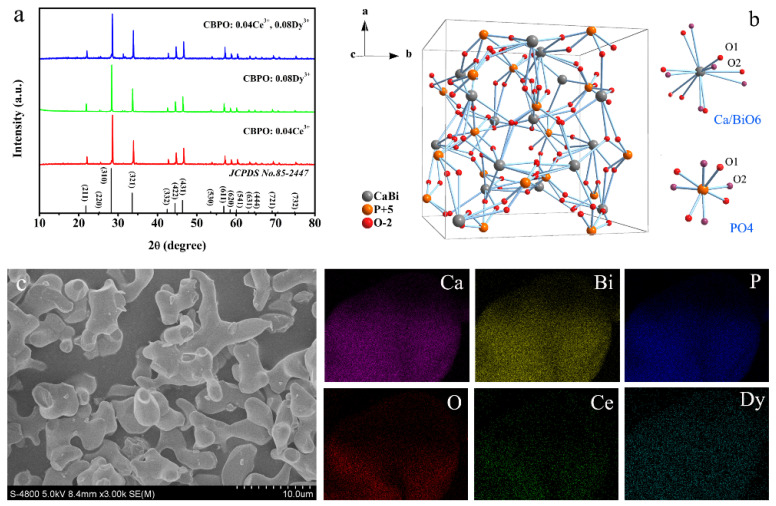
(**a**) XRD patterns of CBPO: 0.04Ce^3+^, CBPO: 0.08Dy^3+^, CBPO: 0.04Ce^3+^, 0.08Dy^3+^ phosphors and the standard for CBPO compounds; (**b**) A structure diagram of CBPO; (**c**) SEM image and the elemental mapping of CBPO: 0.04Ce^3+^, 0.08Dy^3+^ phosphor.

**Figure 2 molecules-28-04967-f002:**
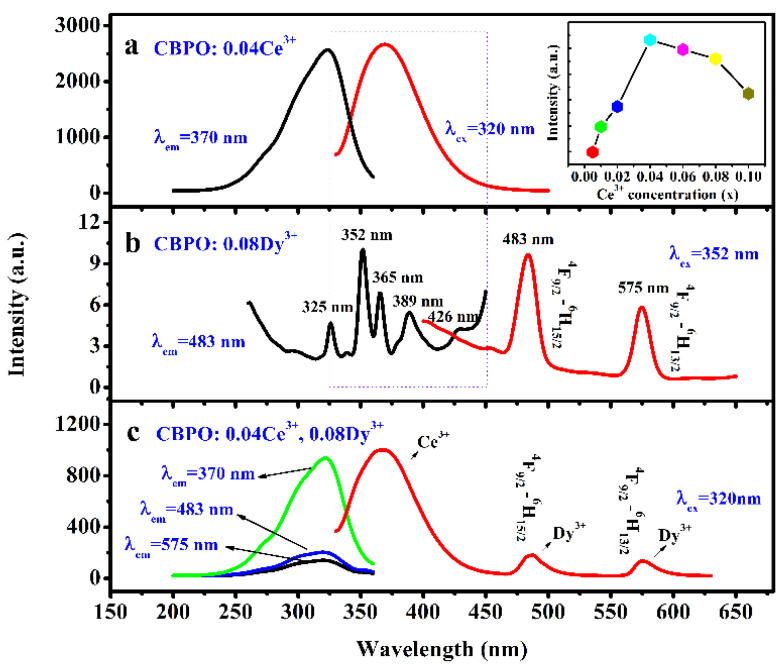
(**a**) The PLE and PL spectra of CBPO: 0.04Ce^3+^, the inset shows the relationship between different doping concentrations of Ce^3+^ ions and luminescence intensity; (**b**) CBPO: 0.08Dy^3+^; (**c**) CBPO: 0.04Ce^3+^, 0.08Dy^3+^.

**Figure 3 molecules-28-04967-f003:**
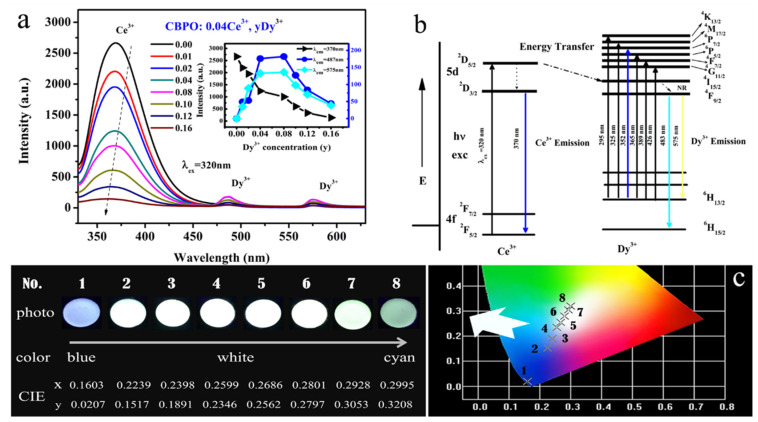
(**a**) The PL spectra of a series of CBPO: 0.04Ce^3+^, yDy^3+^ (y = 0.00, 0.01, 0.02, 0.04, 0.08, 0.10, 0.12, 0.16) phosphors, the inset shows the variation of emission intensity as function of doped Dy^3+^ dopant concentration; (**b**) Energy transfer diagram of CBPO: Ce^3+^, Dy^3+^; (**c**) CIE diagram of the CBPO: 0.04Ce^3+^, yDy^3+^ (y = 0.00, 0.01, 0.02, 0.04, 0.08, 0.10, 0.12, 0.16) phosphors and digital photos of these phosphors.

**Figure 4 molecules-28-04967-f004:**
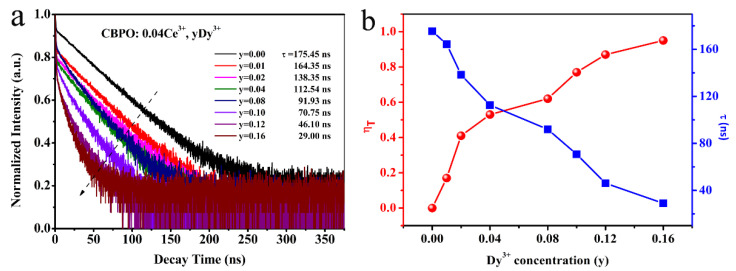
(**a**) The decay curves of CBPO: 0.04Ce^3+^, yDy^3+^ phosphors; (**b**) The dependence of the luminescent lifetime of Ce^3+^ and the η*_T_* on the doped Dy^3+^ concentration.

**Figure 5 molecules-28-04967-f005:**
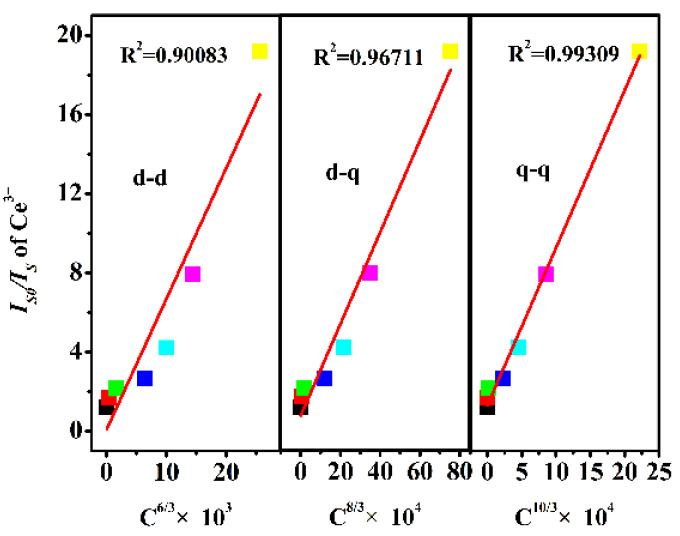
Dependence of *I_S0_/I_S_* of Ce^3+^ on *C^6/3^*, *C^8/3^* and *C^10/3^*.

**Figure 6 molecules-28-04967-f006:**
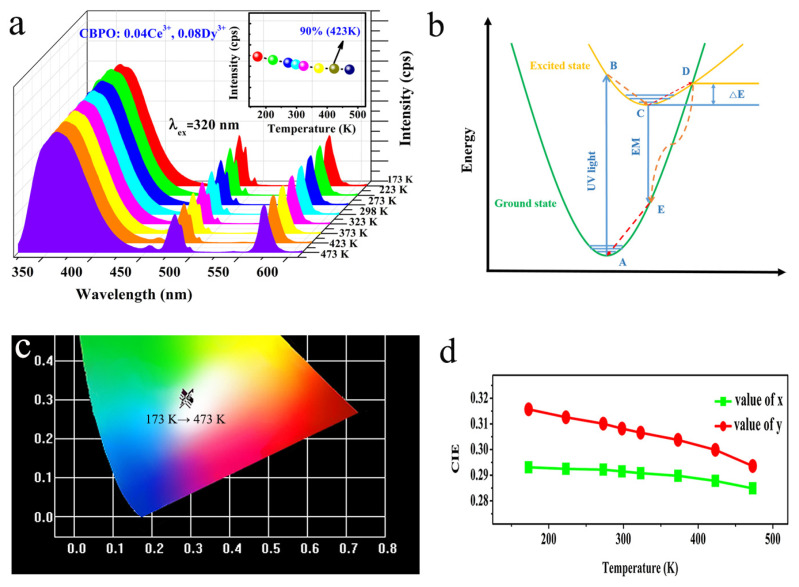
(**a**) Selected temperature-dependent emission spectra of CBPO: 0.04Ce^3+^, 0.08Dy^3+^ phosphor at various temperatures (T = 173 K, 223 K, 273 K, 298 K, 323 K, 373 K, 423 K, 473 K); (**b**) The schematic of configuration coordinate diagram; (**c**) CIE position of the CBPO: 0.04Ce^3+^, 0.08Dy^3+^ phosphor at various temperatures; (**d**) Variation of CIE coordinate at different temperatures.

**Table 1 molecules-28-04967-t001:** CIE coordinates of the samples at different working temperatures.

T (K)	CIE (x)	CIE (y)
173	0.2931	0.3157
223	0.2925	0.3126
273	0.2922	0.3101
298	0.2915	0.3082
323	0.2908	0.3066
373	0.2898	0.3038
423	0.2878	0.2999
473	0.2849	0.2936

## Data Availability

Data are contained within the article.

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
