# Peer review of "Improvement of Luminescence Properties of Eulytite Single-Phase White Emitting Ca3Bi (PO4)3: Ce3+/Dy3+ Phosphor"

_molecules, 2023, doi:10.3390/molecules28134967_

Round 1

Reviewer 1 Report

Comments:

Title: “Improvement of luminescence properties of eulytite single-phase white emitting Ca3Bi (PO4)3: Ce3+/Dy3+ phosphor”

Journal: Molecules (ISSN 1420-3049)

Manuscript Number: molecules-2434356

In a recent study, Xu et al described the development of a novel approach to single-phase phosphors as light-emitting diodes (LEDs). The structures, morphology, photoluminescence, thermal stability, and luminescence mechanism of a variety of Ca3Bi (PO4)3 (CBPO): Ce3+/Dy3+ phosphors were investigated. They have shown that CBPO material can be a potential use for white light phosphor under near-UV excitation at the optimized concentration of Ce3+ and Dy3+. In my opinion, the present paper does not provide any novel contributions. It is therefore recommended that this manuscript not be published until extensive and significant revisions have been made.

A major revision with fresh submission is recommended.

Several points need to be addressed by the authors.

1.     Several similar works have been published in the past, but the author did not cite the most recent ones. It is necessary for the author to conduct a literature review.

2.     There have been similar works published, so it is necessary for the authors to explain what makes their work unique.

https://link.springer.com/article/10.1007/s10854-021-07162-0

https://www.sciencedirect.com/science/article/abs/pii/S0272884214001643?via%3Dihub

https://pubs.rsc.org/en/content/articlelanding/2016/ra/c6ra09926b/unauth

3.     The manuscript lacks scientific insight. It is, therefore, necessary to provide additional information in introduction.

4.     A detailed explanation of the synthesis and chemistry of white phosphors with a reasonable proportion of dopant is required.

5.     The SEM shows very big solid CBPO particles sizes of a few micrometers, so how are the this big particles have been luminescence (moreover without defect)? Could you please include the small angle XRD of CBPO if possible? Additionally, a transmission electron micrograph of the CBPO is required.

6.     Additionally, the author did not provide a proper explanation of the PLE and PL spectra of CBPO. In addition, we are observing many peaks between 250 nm to 450 but there is no proper explanation for these peaks. Therefore, I strongly encourage the author to provide a detailed explanation of all of the physical and chemical characterizations described in the manuscript.

7.     It is also necessary for the author to demonstrate the stability of the material by XRD SEM analysis.

.

Author Response

We would like to thank the reviewers for their valuable suggestions and comments. Those comments are all valuable and very helpful for revising and improving our paper, as well as the important guiding significance to our researches. According to the reviewers’ comments, we carefully checked and revised the entire manuscript. All the questions are addressed in the file “responses to decision letter”. Details of the responses to reviewers' comments and amendments are attached.

Reviewer 2 Report

The manuscript entitled “Improvement of luminescence properties of eulytite single- phase white emitting Ca3Bi(PO4)3:Ce3+/Dy3+ phosphor” was interesting and well-written by the authors. All the necessary characterizations were done and the discussion has been supported by the experimental data. The authors clearly explained the luminescence properties of Ca3Bi(PO4)3:0.04Ce3+/yDy3+ phosphor. However, quantum efficiency is one of the key parameters to assess the practical applicability of phosphor materials. So, the authors need to provide the quantum efficiency of the optimum sample. Also, in the conclusion, the authors mentioned that “A series of CBPO: xCe3+ and CBPO: 0.04Ce3+, yDy3+ phosphors have been synthesized”, but I didn’t find information about different concentrations of Ce3+ ions doped CBPO. In the manuscript, the authors used a fixed concentration (0.04) for Ce3+ ions. It would be better if the authors provide the related data in the manuscript otherwise they need to correct that sentence. 

Author Response

(The authors gave the same response as above.)

Round 2

Reviewer 1 Report

.

ok